# The Combined Effects of the Most Important Dietary Patterns on the Incidence and Prevalence of Chronic Renal Failure: Results from the US National Health and Nutrition Examination Survey and Mendelian Analyses

**DOI:** 10.3390/nu16142248

**Published:** 2024-07-12

**Authors:** Yanqiu Huang, Shiyu Xu, Tingya Wan, Xiaoyu Wang, Shuo Jiang, Wentao Shi, Shuai Ma, Hui Wang

**Affiliations:** 1Department of Nephrology, Shanghai Ninth People’s Hospital, Shanghai Jiao Tong University School of Medicine, Shanghai 200011, China; huangyanqiu@sjtu.edu.cn; 2School of Public Health, Shanghai Jiao Tong University School of Medicine, Shanghai 200025, China; yoouugg@sjtu.edu.cn (S.X.); tinya2000@163.com (T.W.); hooleejnn@sjtu.edu.cn (S.J.); 3Department of Gastroenterology, Shanghai Renji Hospital, Shanghai Jiao Tong University School of Medicine, Shanghai 200127, China; wangxiaoyu6713@126.com; 4Clinical Research Unit, Shanghai Ninth People’s Hospital, Shanghai Jiao Tong University School of Medicine, Shanghai 200011, China; shiwentaotobe@163.com

**Keywords:** dietary patterns, nutrients, chronic kidney disease, dialysis, NHANES, Mendelian randomization

## Abstract

Background: We aimed to comprehensively assess the relationship of specific dietary patterns and various nutrients with chronic kidney disease (CKD) and its progression. Methods: The observational study data were from the NHANES 2005–2020. We calculated four dietary pattern scores (healthy eating index 2020 (HEI-2020), dietary inflammatory index (DII), alternative mediterranean diet (aMed), and dietary approaches to stop hypertension (DASH)) and the intakes of various nutrients and defined CKD, CKD—very high risk, and kidney dialysis. Associations between dietary patterns and nutrients and disease were assessed by means of two logistic regression models. Two-sample MR was performed with various food and nutrients as the exposure and CKD, kidney dialysis as the outcome. Sensitivity analyses were conducted to verify the reliability of the results. Results: A total of 25,167 participants were included in the analyses, of whom 4161 had CKD. HEI-2020, aMed, and DASH were significantly negatively associated with CKD and CKD—very high risk at higher quartiles, while DII was significantly positively associated. A higher intake of vitamins and minerals may reduce the incidence and progression of CKD to varying degrees. The MR results, corrected for false discovery rates, showed that a higher sodium intake was associated with a higher prevalence of CKD (OR: 3.91, 95%CI: 2.55, 5.99). Conclusions: Adhering to the three dietary patterns of HEI-2020, aMed, and DASH and supplementing with vitamins and minerals benefits kidney health.

## 1. Introduction

Chronic kidney disease (CKD) is a growing global health crisis, posing a significant burden in terms of morbidity, mortality, and healthcare costs [1]. It is estimated that over 10% of adults worldwide are affected by CKD [2]. Metabolic diseases like diabetes mellitus and hypertension are key risk factors for CKD’s development and progression [3,4]. Diabetes mellitus leads to kidney injury through multiple pathways, including inflammation and the activation of pro-fibrotic processes [5,6]. Hypertension is also a metabolic disease in a certain sense. Recent studies have suggested a strong link between salt intake, hepatic metabolism, and blood pressure [7,8]. The metabolic rewiring of the kidneys during hypertension highlights the potential for metabolic or dietary interventions to target hypertension-induced kidney disease [9]. Other diet-related factors, such as obesity and dyslipidemia, also play an important role in the pathogenesis of CKD. The treatment of these metabolic diseases relies heavily on dietary interventions. CKD patients frequently encounter complex nutritional disorders, including malnutrition, protein–energy wasting (PEW), and overnutrition leading to obesity [10,11]. Nutritional disorders in CKD heighten the risk of disease progression and worsens metabolic derangements [10].

The management of CKD requires a multidisciplinary approach, with nutrition playing a crucial role [12]. Dietary interventions for CKD patients typically involve the adjustment of their intake of specific nutrients, including protein, sodium, potassium, phosphorus, and others. By understanding nutritional management strategies, healthcare professionals can play a vital role in optimizing the care of and outcomes for individuals with CKD. Observational research has shown that dietary patterns promoting cardiovascular health are associated with reduced mortality [13] and a slower progression of CKD [14,15]. Furthermore, clinical trials support the benefits of whole-diet interventions for CKD, which suggest that dietary patterns targeting the overall diet can have a positive impact on slowing the decline in renal function [16,17]. Instead of solely focusing on the manipulation of individual nutrients, the emphasis has now shifted towards promoting healthy, balanced dietary patterns that can address the multifaceted nutritional and metabolic needs of patients with CKD. This holistic approach recognizes that the complex interplay between various dietary components and their synergistic effects on overall health may be more impactful than the isolated management of specific nutrients.

While substantial evidence links healthy dietary patterns to the primary prevention of conditions like diabetes and hypertension, their role in preventing CKD is less clear. This study aimed to investigate the role of four healthy dietary patterns in the incidence of CKD using data from the National Health and Nutrition Examination Survey (NHANES). Additionally, the study employed Mendelian randomization (MR) analyses to explore the potential causal relationship between these dietary patterns and CKD.

## 2. Materials and Methods

### 2.1. Study Population and Design

Participants in this observational study were drawn from NHANES 2005–2020. NHANES selects nationally representative samples of approximately 5000 individuals per year using multistage complex sampling every two years. Of the 33,385 participants with data on dietary and diagnostic indicators of CKD, those with missing data on lifestyle and body measurements were excluded. To reduce bias, pregnant women (*n* = 292) were also excluded from the analyses, and the final study included 25,167 participants, with a subsample of 906 who had data answering the questionnaire about renal dialysis (Appendix A). All participants provided written informed consent. The NHANES protocol was approved by the Centers for Disease Control (CDC) and Prevention Institutional Review Board.

### 2.2. Dietary Scores

NHANES assessed dietary information using 24 h recall data. Trained interviewers conducted interviews with each participant to collect information on the type and amount of food and beverages consumed by the participant in the past 24 h and to estimate the energy and nutrients consumed from these foods and beverages. Telephone interviews were conducted with all participants after 3–10 days. If dietary information was available on two occasions, averaging was chosen. The nutritional values for dietary information were obtained from the United States Department of Agriculture (USDA) food composition database. Data from the Dietary Interview and Food Pattern Equivalents database files were used to calculate dietary scores [18]. In addition, we included all nutrients associated with dietary patterns as exposures for analysis.

#### 2.2.1. Healthy Eating Index 2020 (HEI-2020)

The HEI-2020 is a dietary quality index developed by the USDA’s Center for Nutrition Policy and Promotion based on the Dietary Guidelines of Americans (DGA) 2020–2025. The thirteen components and scoring criteria of the HEI-2020 are identical to those of the HEI-2015, and although it does not incorporate any changes, the index was renamed to clarify its alignment with the 2020–2025 DGA (Appendix A). Scores range from 0 to 100. Higher scores reflect healthier diets [19].

#### 2.2.2. Dietary Inflammatory Index (DII)

The DII compares the inflammatory potential of individual diets based on 45 pro- and anti-inflammatory food parameters. In NHANES, there are 28 food parameters that can be used for DII calculations (Appendix A). Nutrient- and final energy-adjusted DIIs per 1000 calories consumed were calculated using energy density methods (using naturalized global intakes from 11 populations around the world). A participant’s DII is the sum of each DII. Higher positive DIIs are more pro-inflammatory, while higher negative DIIs are more anti-inflammatory [20].

#### 2.2.3. Alternative Mediterranean Diet (aMed)

The aMed score assesses the adherence to the traditional Mediterranean diet. An aMed score consists of nine components (Appendix A). One point was awarded if the individual consumed above the sex-specific group median and 0 for less than the median, while components that are considered harmful such as a monounsaturated/saturated fat ratio, red meats and their products, poultry, and dairy products were given one point for intake below the median and 0 for intake above the median; for alcohol consumption, one point was given to women for an intake of 5–25 g/day and to men for an intake of 10–50 g/day, indicating that their intake was moderate, and 0 points for consumption outside these ranges. Potatoes were excluded from the NHANES calculations due to differences in preparation methods between the United States and Europe. aMed scores range from 0 to 9, with higher scores indicating a better aMed pattern [21].

#### 2.2.4. Dietary Approaches to Stop Hypertension (DASH)

The DASH dietary pattern is effective in reducing blood pressure, and a total DASH score is calculated based on the nine DASH indices for the nutrients (Appendix A). One point is awarded for meeting the goal for each component, and participants are awarded one point. The pattern score ranges from 0 to 9, where higher scores indicate greater adherence to the DASH pattern [22].

### 2.3. The Diagnosis of CKD Staging and Kidney Dialysis

NHNAES collected urine samples and blood samples from each participant, and we obtained serum creatinine, urinary creatinine, and albumin (urinary albumin creatinine ratios, ACRs) from participants through a central laboratory. The ACR was classified as less than 30 (A1), 30–300 (A2), or more than 300 (A3) mg/g [23]. The eGFR was categorized as G1 (>90), G2 (60–89), G3a (45–59), G3b (30–44), G4 (15–29), and G5 (<15 mL/min/1.73 m^2^) [24]. Patients with CKD were defined if they had an eGFR < 59 or an ACRs > 30. In addition, CKD—very high risk (G3b and A2–A3 or G3a and A3, G4–G5) was defined, as such individuals would be advised to undergo dialysis treatment [25].

In addition to the CKD stages, some participants answered that they had “received dialysis in the last 12 months”, and those who answered “Yes” were defined as being on kidney dialysis. This subsample was included in the sensitivity analysis.

### 2.4. Statistical Analysis

Participants’ characteristics as a whole were stratified by the presence or absence of CKD and expressed as mean (standard error, SE) for continuous variables or frequency (%) for categorical or ordinal variables. All analyses were weighted to account for the complex sampling and national representativeness of NHANES. Logistic regression assessed the associations between dietary pattern scores, various nutrients, and nephropathy status. Dietary pattern scores were categorized by quartiles (Q1, Q2, Q3, Q4), and the various nutrients were categorized by tertiles (T1, T2, T3). In the regression models, Model 1 adjusted for age (continuous, years), gender (male and female), and race (non-Hispanic white, Hispanic, etc.), and based on Model 1, we additionally adjusted for the level of education (less than high school and high school or above), poverty-to-income ratio (PIR) (<1.3, 1.3–3.5, ≥3.5), smoking status (never smoked and smoking), leisure time physical activity (LTPA) (adequate and inadequate) [26], body mass index (continuous, kg/m^2^), and history of diabetes (yes and no). In addition, the total energy intake was adjusted to balance the differences in dietary intake between individuals. Dose–response relationships between dietary pattern scores, various nutrients, and nephropathy status were estimated using a restricted cubic spline (RCS) based on multivariate logistic regression, and 3 knots were established in our model. Risk estimates were adjusted for covariates that were consistent with Model 2. Nonlinearity was estimated in the model based on the *p*-value. Statistical significance was considered to be a *p*-value < 0.05, and all tests were two-tailed. All statistical analyses were performed using R (4.2.3, The R Foundation, Vienna, Austria).

### 2.5. MR Design

The detailed description of the overall MR study design and assumptions is shown in Appendix A. A properly constructed MR is based on three assumptions: (i) genetic variants are linked to risk factors; (ii) genetic variants are not influenced by confounding factors; and (iii) genetic variants impact outcomes solely through risk factors [27]. 

#### 2.5.1. Data Source and Selection of IVs

In this MR analysis, we utilized instrumental variables to investigate the relationship between potential diet-related risk factors and CKD or dialysis individually. Sixty-five diet-related predominant modifiable risk factors were grouped into eleven categories: (i) vegetables and fruits; (ii) grains; (iii) nuts and legumes; (iv) total meat; (v) ultra-processed food; (vi) beverages and juice; (vii) alcohol; (viii) tea and coffee; (ix) dairy products; (x) fatty acids; and (xi) micro-elements. Data were obtained from genome-wide association studies (GWASs) that are accessible to the public. We identified the significant genetic differences linked to each characteristic with *p*  <  5 × 10^−8^. Since a limited number of single nucleotide polymorphisms (SNPs) associated with CKD or dialysis could be obtained at the level of the standard genome-wide significance of risk traits like white rice, lamb intake, and so on, the genetic instruments were set between *p*  <  5 × 10^−8^ and *p*  <  1 × 10^−5^. Appendix A contains more detailed information. A linkage disequilibrium (LD) test was conducted to eliminate linked SNPs with r^2^ < 0.01 within a 5000-kilobase-pair window [28]. Finally, the F-statistic was computed individually for each SNP, and those with F < 10 were removed to ensure the reliability of the genetic instruments for each potential risk factor and avoid weak instrument bias [29].

As most of the exposure factors came from UK Biobank, we excluded the GWAS data of CKD and dialysis based on the UK Biobank to reduce overlap. Summary-level genetic data for CKD (ncase = 41,395, ncontrol = 439,303) were extracted from the CKDGen Consortium (https://ckdgen.imbi.uni-freiburg.de/, accessed on 10 December 2023) [30], and the data for dialysis (ncase = 1004, ncontrol = 363,177) were obtained from the FinnGen consortium release 9 (https://www.finngen.fi/fi/, accessed on 15 December 2023) [31]. Since all analyses herein were based on publicly available summary data, no ethical approval from the ethics committee or institutional review board was required for this study.

#### 2.5.2. MR Statistical Analysis

The selection of instrumental variants, MR analyses, and result visualization were performed using R and R packages, which mainly contributed to TwoSampleMR and MRPRESSO. We utilized the random-effect inverse-variance-weighted (IVW) method as the primary statistical model, because it was reported as the most often best-performing method when all selected SNPs were valid IVs [32,33]. The random IVW method may ease the exclusion restriction assumption by enabling all SNPs to exhibit random horizontal polymorphic effects [34]. Nevertheless, the bias risk can still exist if the instrument’s SNPs exhibit horizontal pleiotropy, impacting the outcome through causal pathways beyond the exposure and breaching assumptions of the instrumental variable. Hence, we conducted sensitivity analyses by comparing IVW outcomes with other well-known MR techniques that are generally resistant to horizontal pleiotropy, albeit with a slightly lower statistical power. These methods include the weighted median estimator [35], which can yield impartial causal effects with a minimum of 50% valid SNPs chosen, and MR–Egger regression, which conducts a weighted linear regression and produces a reliable causal estimate despite all SNPs in an instrument being invalid due to pleiotropy [36]. But the significant influence of outlying genetic variables may contribute to the MR–Egger model’s low statistical ability. We also applied a weighted mode to estimate the causal impact of the subset with the most SNPs by grouping the SNPs based on their resemblance to causal effects, and simple mode, although less powerful than IVW, can offer protection against pleiotropy [37,38].

Ultimately, we further performed a leave-one-out sensitivity test and MR–Egger intercept test to assess the impact of outlier and pleiotropic SNPs on causal assessments [39]. The heterogeneity and pleiotropy of individual SNPs were evaluated using Cochran Q statistics and MR-PRESSO. In order to deal with multiple hypothesis testing, we applied the false discovery rate (FDR)-adjusted *p*-values (*p <* 0.05) in both the primary IVW MR analyses and sensitive analyses, following the sequential *p*-values method suggested by Benjamini and Hochberg [40]. The power calculation for this MR study was achieved using an online web tool at https://sb452.shinyapps.io/power/. All statistical analyses were performed using R (4.2.3, The R Foundation, Vienna, Austria).

## 3. Results

### 3.1. Population Characteristics

We included 25,167 participants from NHANES 2005–2020, of whom 4161 (16.5%) were diagnosed with CKD (Table 1 and Appendix A). Patients with CKD were older (62.3 years), had a higher proportion of females (53.3%), higher BMIs (30.5 kg/m^2^), and a higher proportion of smokers (49.5%) compared to the non-CKD population, who had a lower education level, their PIR was lower, and the proportion of LTPA was lower. In the dietary patterns, there were significant differences between the two groups in the DII and DASH scores (*p* < 0.001, *p* = 0.013). Except for retinol, VA, α-carotene, β-carotene, and β-cryptoxanthin, there were significant differences in the intake of all other nutrients between the two groups.

### 3.2. Association between Dietary Scores and CKD and CKD—Very High Risk

The RCS plots (Appendix A) evaluate the nonlinear relationship between dietary scores and CKD and CKD—very high risk. The results showed that the HEI-2020 and aMed scores showed a significant nonlinear negative association with CKD. As shown in Table 2, HEI-2020 and aMed significantly reduced the CKD incidence by 22% and 19% at Q4 intake and reduced the CKD—very high risk incidence by 34% (Q3), 54% (Q4), 39% (Q3), and 59% (Q4), respectively. DASH was significantly negatively associated with both outcomes only in Model 1 (*p* < 0.001, *p* = 0.012). In contrast, the DII was significantly positively associated with both outcomes at Q3 or Q4 (*p* < 0.001), and the results of the two models were largely consistent.

### 3.3. Subgroup Analysis of Age and Gender

Consistent with the total population, HEI-2020 and aMed have a significant barbed curve association with CKD in males and age ≥ 45. However, there was no nonlinear relationship among females and young adults. In addition, among adults over 45, DASH has an inverted-N-shaped curve with CKD. There was no significant nonlinear relationship between dietary score and CKD—very high risk (Figure 1 and Appendix A).

### 3.4. Associations between Various Nutrient Intakes and CKD and CKD—Very High Risk

As shown in Figure 2, retinol, VB (except VB12), VE, VK, total folate, total choline, calcium, phosphorus, magnesium, iron, zinc, copper, and potassium were significantly and negatively associated with CKD at the T2 or T3 of intake. Inconsistent with the CKD association, VB1, VB2, total choline, calcium, iron, and potassium were statistically significantly negatively associated with CKD—very high risk only at T3 intake. In addition, VB12, VD, sodium, and selenium were significantly negatively associated with CKD only at T3 intake (OR: 0.82, 95%CI: 0.72, 0.93; OR: 0.82, 95%CI: 0.71, 0.94; OR: 0.74, 95%CI: 0.66, 0.84; OR: 0.83, 95%CI: 0.72, 0.95). β-carotene was significantly negatively associated with CKD—very high risk. Appendix A demonstrate the nonlinear associations between macronutrients, vitamins, minerals, and fatty acids with CKD and CKD—very high risk. Appendix A presents the association between macronutrient and fatty acid intake and two outcomes by means of logistic regression.

### 3.5. Mendelian Randomization Analysis of Various Foods and Nutrients with CKD, Dialysis

Figure 3 summarizes the discovery results for CKD in the CKDGen consortium and dialysis in the FinnGen consortium. Regarding the diet-related factors investigated, genetically predicted higher unsalted nut intake, broad bean intake, Indian snack intake, fizzy drink intake, and fortified wine intake were significantly associated with a decreased risk of CKD. The ORs (95% CI) of the IVW method were 0.64 (0.43, 0.94; *p*  =  0.024), 0.42 (0.18, 0.99; *p* = 0.047), 0.30 (0.10, 0.97; *p* = 0.044), 0.76 (0.58, 0.99; *p* = 0.039), and 0.25 (0.08, 0.75; *p* = 0.014) for genetically predicted 1-SD increases, respectively, indicating that these traits might act as protective factors. In addition, there was a positive association between UNa/UCr, which reflects the sodium intake and an increased risk of CKD (IVW: OR = 3.91, *p* < 0.0001) for a 1-SD increase, which suggested that sodium intake could be a strong indicator of an increased risk of CKD. After adjusting for FDR, only UNa/UCr showed a significantly positive association with the risk of CKD. We also found that genetically predicted white rice intake (IVW: OR = 0.02, *p* = 0.036), broad bean intake (IVW: OR = 0.01, *p* = 0.034), and non-oily fish intake (IVW: OR = 0.05, *p* = 0.021) had significant relationships with a lower risk of dialysis, while VB12 (IVW: OR = 3.43, *p* = 0.006) intake increased the risk for dialysis. However, after adjusting for FDR, we observed no evidence of causal relationships of diet-related traits with dialysis.

### 3.6. Sensitivity Analysis

We assessed the association between dietary scores and renal dialysis by means of RCS and the logistic regression model (Figure 4). The HEI-2020 and aMed scores all had significant barbed-type associations with kidney dialysis. The aMed scores significantly reduced kidney dialysis occurrence at ≥6 points (OR: 0.38, 95%CI: 0.22, 0.68). A DII at ≥1.45 points significantly increased kidney dialysis incidence by 1.82-fold (OR: 0.38, 95%CI: 1.55, 5.13). In addition to the IVW method, the MR analysis used the sensitive analysis including the weighted median, MR–Egger, weighted mode, simple mode yielded, and leave-one-out sensitivity test, which yielded a similar pattern of effects (Appendix A).

## 4. Discussion

In this large, nationally representative study of US adults, we found that healthier dietary patterns, characterized by a higher adherence to the HEI-2020 and aMed scores were significantly associated with a lower incidence of CKD and CKD—very high risk. Conversely, a pro-inflammatory diet, as reflected by a higher DII score, was associated with an increased risk of these kidney outcomes. Notably, the associations between dietary scores and CKD demonstrated nonlinear relationships, with the most significant benefits observed at the highest quartiles of HEI-2020 and aMed scores.

The inverse associations between the HEI-2020, aMed scores, and CKD risk observed in our study are consistent with previous research. A meta-analysis of eight studies found that a higher adherence to a healthy diet was associated with a 30% lower incidence of CKD [41]. Similarly, a recent study based on the Chronic Renal Insufficiency Cohorts reported that higher aMed and DASH scores can the reduce risk of CKD progression and all-cause mortality [42]. The protective effects of these dietary patterns on kidney health can be attributed to their emphasis on plant-based foods, healthy fats, and a limited intake of refined carbohydrates and processed meats, all of which have been linked to improved metabolic and inflammatory profiles [43,44,45]. Regarding the DASH diet, our study only found a significant negative association with CKD and CKD—very high risk in the minimally adjusted model (Model 1), but not in the fully adjusted model (Model 2). This discrepancy may be due to the complex interplay between the individual components of the DASH diet and their effects on kidney health. The DASH diet is characterized by a high intake of fruits, vegetables, whole grains, and low-fat dairy, as well as a low intake of sodium, red and processed meats, and added sugars [15]. While the high intake of fruits, vegetables, and low-fat dairy may confer beneficial effects on kidney function through their provision of essential vitamins, minerals, and antioxidants, the potential favorable impact of the DASH diet may be attenuated by the complex relationships between its individual dietary components and CKD risk [46]. For instance, while reducing one’s sodium intake is generally recommended for kidney health [47,48], the DASH diet also emphasizes the consumption of dairy products, which may have a more nuanced effect on CKD risk depending on factors such as calcium and phosphorus intake [49,50]. Further research is needed to elucidate the specific mechanisms by which the DASH diet and its individual components influence kidney outcomes.

In contrast, the positive association between the DII score and CKD risk observed in our study aligns with previous evidence. One cohort study has reported a 12% increased risk of CKD with higher DII scores [51]. Chronic low-grade inflammation is a well-recognized contributor to the development and progression of CKD, and dietary factors that promote inflammation, such as processed and red meats, refined carbohydrates, and certain oils, may exacerbate this inflammatory process [52,53,54,55]. The DII is a comprehensive dietary index that captures the pro-inflammatory or anti-inflammatory potential of an individual’s diet, and its association with an increased CKD risk underscores the importance of following dietary patterns that minimize inflammation. The pro-inflammatory effects of these dietary components can lead to endothelial dysfunction, oxidative stress, and impaired renal hemodynamics, ultimately contributing to the initiation and progression of CKD [56,57,58,59]. This suggests that dietary interventions aimed at reducing inflammation may have the potential to mitigate the risk and progression of CKD.

The subgroup analyses in our study revealed that the associations between dietary scores and CKD risk were more pronounced among older adults and males. This suggests that the beneficial effects of healthy dietary patterns on kidney health may be particularly relevant for these populations, who often have a higher risk of CKD [60]. Among younger adults and females, the lack of a significant nonlinear relationship between dietary scores and CKD outcomes may indicate that other factors, such as genetic predisposition, physical activity, and socioeconomic status, play a more prominent role in determining the CKD risk in these groups [61,62].

Regarding the associations between individual nutrient intake and CKD outcomes, our results provide valuable insights into the potential mechanisms underlying the observed dietary patterns–CKD relationships. We found that higher intakes of several vitamins (A, B-complex, E, K), minerals (calcium, phosphorus, magnesium, iron, zinc, copper, potassium), and antioxidants (retinol, β-carotene) were associated with a lower risk of CKD. These nutrients have been shown to exert beneficial effects on kidney function through various pathways, such as reducing oxidative stress, improving endothelial function, and modulating inflammation [56,63,64]. For example, retinol and its precursor, β-carotene, have been found to possess anti-inflammatory and anti-fibrotic properties, which may help prevent the progression of CKD [65,66]. Trasino SE et al. [67] found that retinoic acid ameliorated the inflammatory response mediated by the TLR4/NF-κB pathway in a CKD animal model. Vitamin E, a potent antioxidant, can protect the kidney from oxidative damage and improve the glomerular filtration rate. In 2021, a small randomized controlled trial reported that daily supplementation with 400 mg of tocotrienol-rich vitamin E significantly improved the renal function in the CKD patients [68]. B-complex vitamins, such as folate and vitamin B6, play crucial roles in homocysteine metabolism, and their deficiency has been linked to an increased incidence of CKD [69,70]. The CSPPT Renal Sub-study [71] reported that folate supplementation was associated with a reduced risk of CKD progression in a hypertensive population. However, in patients with diabetic nephropathy, high-dose B-vitamin supplementation (including folate and vitamin B6) resulted in a greater decline in the glomerular filtration rate over the study period [72]. More studies are still needed on the role of vitamin B in the progression of CKD. Minerals like calcium, phosphorus, and magnesium are essential for maintaining proper bone and mineral metabolism, and their imbalances have been associated with the development and progression of CKD [73,74]. Potassium, a key electrolyte, is crucial for regulating an individual’s fluid balance and maintaining optimal kidney function [75]. Our study found that a lack of potassium intake increased the prevalence of CKD. Due to the prevalence of hyperkalemia in individuals with CKD, dietary potassium restriction is commonly recommended. However, as impaired cardiovascular function can contribute to kidney damage [76], a potassium-restricted diet may potentially be detrimental for patients with CKD.

The MR analysis in our study further supports the potential causal relationships between dietary factors and kidney outcomes. We found that genetically predicted higher intakes of unsalted nuts, broad beans, and certain beverages (fizzy drinks, fortified wines) were associated with a lower incidence of CKD, while a higher sodium intake (reflected by the UNa/UCr ratio) was associated with an increased risk of dialysis. These findings align with previous observational and experimental research highlighting the beneficial effects of nut consumption, legumes, and limiting sodium intake on kidney health [77,78,79,80]. The inverse associations observed between nut and legume consumption and CKD risk may be attributed to their rich content of various nutrients, such as healthy fats, fiber, magnesium, potassium, and antioxidants, which have been shown to exert protective effects on kidney function [81]. Conversely, the increased prevalence of dialysis associated with a higher sodium intake is consistent with the well-established role of excess dietary sodium in promoting hypertension, a major risk factor for CKD progression [47,48].

It is worth noting that the MR analysis did not consistently identify significant causal relationships between all the examined dietary factors and the kidney outcomes, likely due to the limited statistical power and potential pleiotropic effects of some genetic variants used as proxies for dietary exposures. Nonetheless, the overall pattern of results from our observational and MR analyses provides compelling evidence for the role of diet in the prevention and management of CKD.

The strengths of our study include the large, nationally representative sample, the comprehensive assessment of dietary patterns and nutrient intakes, and the use of advanced statistical techniques, such as restricted cubic splines and Mendelian randomization, to examine the complex and potentially nonlinear relationships between diet and kidney outcomes. However, several limitations should be considered. First, the cross-sectional nature of the observational analyses precludes the establishment of temporal relationships and causal inferences. Second, the MR analysis was limited by the availability and quality of genetic instruments for dietary exposures, and the potential for pleiotropy cannot be entirely ruled out.

## 5. Conclusions

In conclusion, our findings demonstrate that healthier dietary patterns, characterized by a higher adherence to the HEI-2020 and aMed scores, are associated with a lower incidence of CKD and CKD—very high risk, particularly among older adults and males. Conversely, a pro-inflammatory diet, as reflected by a higher DII score, is associated with an increased incidence of these kidney outcomes. The beneficial effects of specific nutrient intakes and the potential causal relationships between dietary factors and kidney health, as suggested by the MR analysis, provide further support for the crucial role of diet in the prevention and management of CKD. These results underscore the importance of promoting healthy dietary patterns as a key strategy for kidney disease prevention and care.

## Figures and Tables

**Figure 1 nutrients-16-02248-f001:**
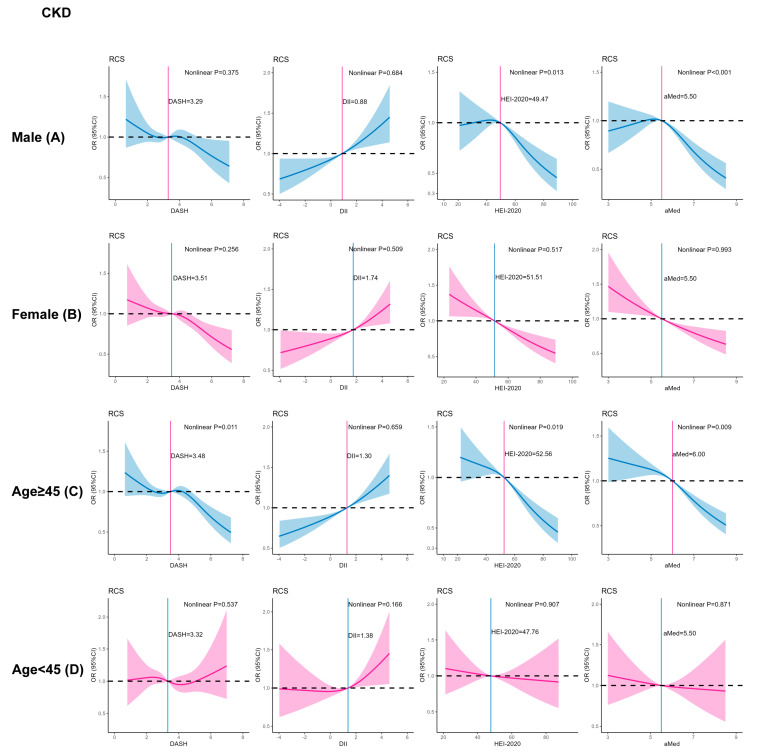
RCS plots of the association between four dietary scores and CKD in gender and age subgroups. Model: age, gender, race/ethnicity, body mass index, smoking status, poverty status, education levels, alcohol consumption, leisure time physical activity, history of diabetes. *p*-values less than 0.05 (*p*-value < 0.05) were considered significant. Abbreviations: CKD, chronic kidney disease; HEI, healthy eating index; DII, Dietary Inflammation Index; aMed, Alternate Mediterranean Diet; DASH, dietary approaches to stop hypertension; N, number.

**Figure 2 nutrients-16-02248-f002:**
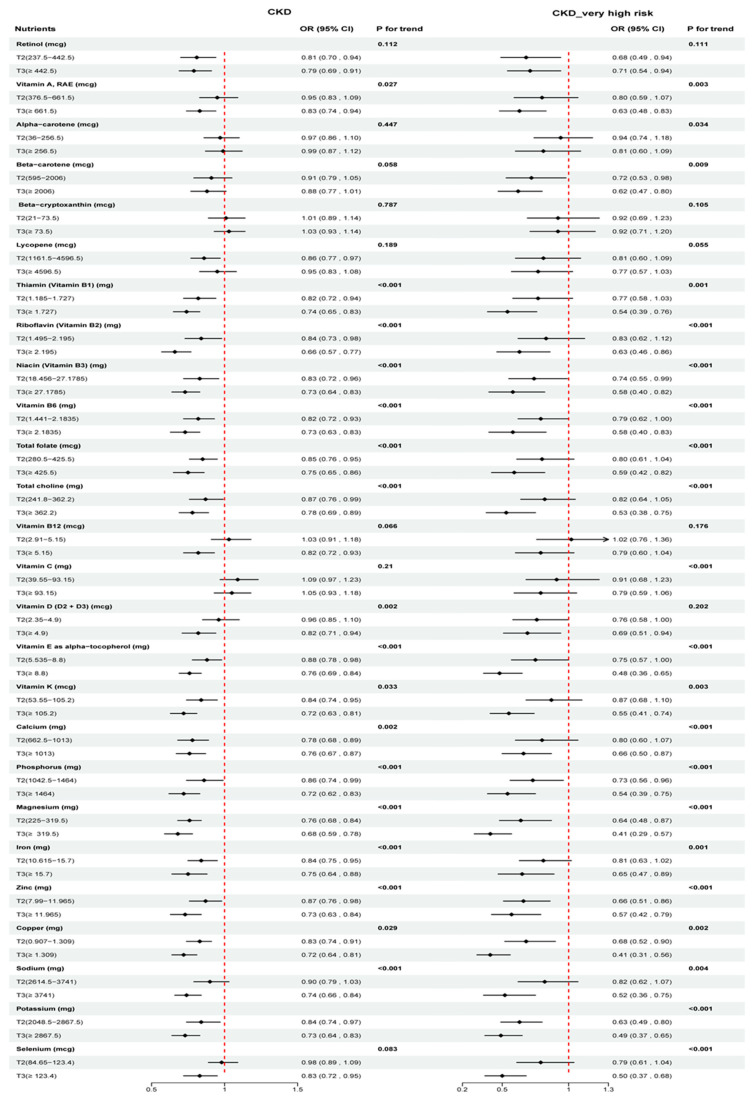
Associations between vitamins, minerals, and CKD/CKD—very high risk in logistic regression models. Model: age, gender, race/ethnicity, body mass index, smoking status, poverty status, education levels, alcohol consumption, leisure time physical activity, history of diabetes.

**Figure 3 nutrients-16-02248-f003:**
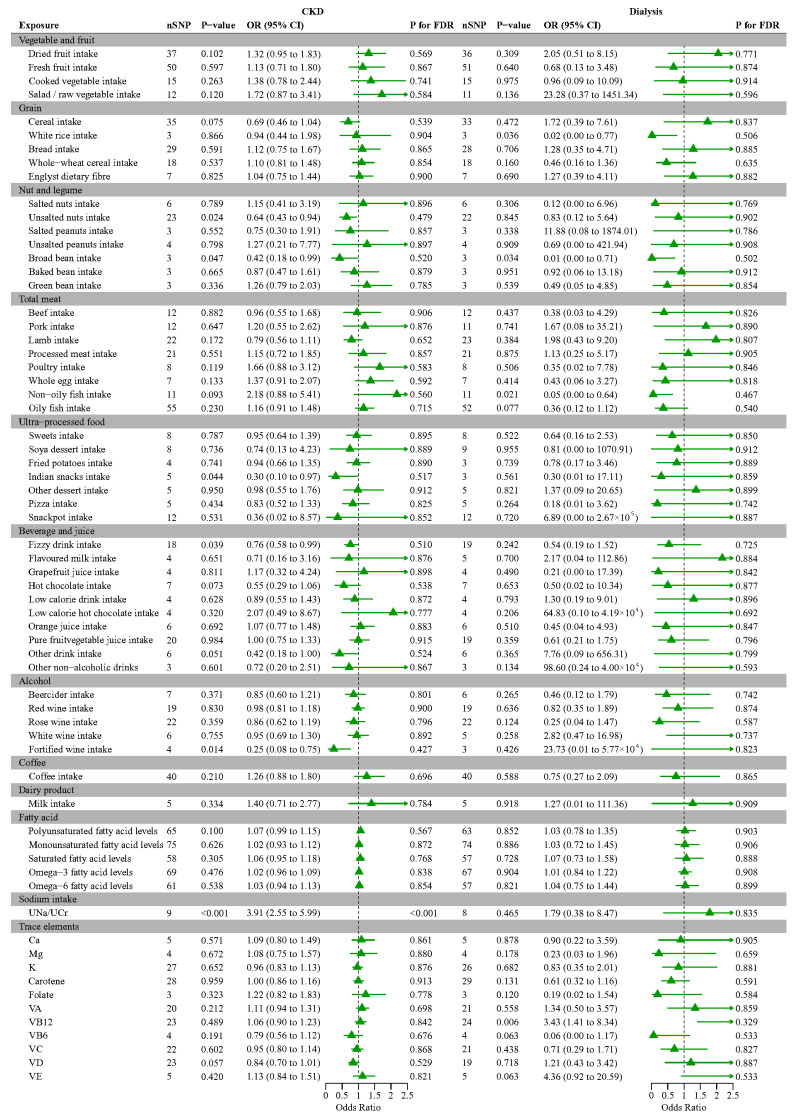
MR analysis of modifiable diet-related risk factors and CKD/dialysis. Abbreviations: CKD, chronic kidney disease; OR, odd ratio; CI, confidence interval.

**Figure 4 nutrients-16-02248-f004:**
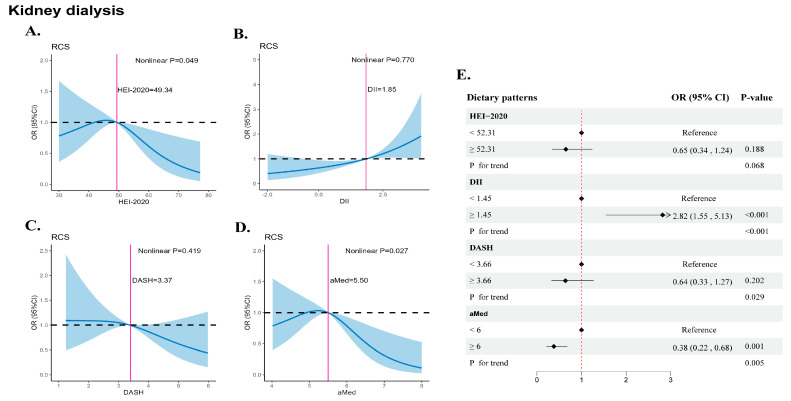
Associations between dietary scores and kidney dialysis. (**A**) HEI-2020, (**B**) DII, (**C**) DASH, (**D**) aMed, (**E**) Dietary pattern scores for dichotomous categories. Model: age, gender, race/ethnicity, body mass index, smoking status, poverty status, education levels, alcohol consumption, leisure time physical activity, history of diabetes.

**Table 1 nutrients-16-02248-t001:** Baseline characteristics: NHANES (2005 to 2020).

Characteristics	Total (N = 25,167)	Non-CKD (N = 21,006)	CKD (N = 4161)	*p*-Value
**Age, years, mean (SD)**	49.2 (17.8)	46.6 (16.9)	62.3 (16.6)	<0.001
**A** **lcohol consumption** **, gram, mean (SD)**	25.1 (34.9)	26.4 (35.4)	18.6 (31.5)	<0.001
**Body Mass Index, kg/m^2^** **, mean (SD)**	29.4 (7.1)	29.2 (7.0)	30.5 (7.5)	<0.001
**eGFR, mL/min/1.73 m^2^, mean (SD)**	96.2 (24.3)	100.6 (20.0)	73.5 (30.6)	<0.001
**Albumin creatinine ratio, mg/g, mean (SD)**	42.9 (338.9)	8.1 (5.6)	810.9 (810.9)	<0.001
**Gender, *n* (%)**				0.025
Male	12,150 (48.3)	10,207 (48.6)	1943 (46.7)	
Female	13,017 (51.7)	10,799 (51.4)	2218 (53.3)	
**Ethnicity, *n* (%)**				<0.001
Mexican American	3549 (14.1)	3034 (14.4)	515 (12.4)	
Other Hispanic	2509 (10.0)	2178 (10.4)	331 (8.0)	
Non-Hispanic White	11,035 (43.9)	9018 (42.9)	2017 (48.5)	
Non-Hispanic Black	5309 (21.1)	4352 (20.7)	957 (23.0)	
Other Race—Including Multi-Racial	2765 (11.0)	2424 (11.5)	341 (8.2)	
**Education level, *n* (%)**				<0.001
Less than high school	5476 (21.8)	4323 (20.6)	1153 (27.7)	
High school or above	19,691 (78.2)	16,683 (79.4)	3008 (72.3)	
**Poverty status, *n* (%)**				<0.001
≤1.30	7797 (31.0)	6402 (30.5)	1395 (33.5)	
1.30–3.50	9441 (37.5)	7692 (36.6)	1749 (42.0)	
>3.50	7929 (31.5)	6912 (32.9)	1017 (24.4)	
**Smoking status, *n* (%)**				<0.001
Yes	10,915 (43.4)	8857 (42.2)	2058 (49.5)	
No	14,252 (56.63)	12,149 (57.8)	2103 (50.5)	
**Leisure time physical activity, *n* (%)**				<0.001
Adequate	8606 (34.2)	7697 (36.6)	909 (21.9)	
Inadequate	16,561 (65.8)	13,309 (63.4)	3252 (78.2)	
History of diabetes, *n* (%)				<0.001
Yes	3267 (13.0)	1911 (9.1)	1356 (32.6)	
No	21,900 (87.0)	19,095 (90.9)	2805 (67.4)	
**History of hypertension, *n* (%)**				<0.001
Yes	7011 (27.9)	4983 (23.7)	2028 (48.7)	
No	12,581 (50.0)	11,422 (54.4)	1159 (28.9)	
Unknown or Missing	5575 (22.2)	4601 (21.9)	974 (23.4)	
**History of cardiovascular disease, *n* (%)**				<0.001
Yes	2048 (8.1)	1164 (5.5)	884 (21.2)	
No	17,181 (68.3)	14,901 (70.9)	2280 (54.8)	
Unknown or Missing	5938 (23.6)	4941 (23.5)	997 (24.0)	

Note: *p*-values less than 0.05 (*p*-value < 0.05) were considered significant. Abbreviations: CKD, chronic kidney disease; N, number; SD, Standard Deviation.

**Table 2 nutrients-16-02248-t002:** Analysis of the association between dietary patterns and chronic kidney disease (CKD) and CKD—very high risk.

Dietary Patterns	CKD						CKD—Very High Risk					
	Model 1			Model 2			Model 1			Model 2		
	OR	95%CI	*p*-Value	OR	95%CI	*p*-Value	OR	95%CI	*p*-Value	OR	95%CI	*p*-Value
**HEI-2020**												
Q1 (<42.523)	**Reference**			**Reference**			**Reference**			**Reference**		
Q2 (42.523–50.483)	0.88	(0.76, 1.02)	0.082	0.93	(0.80, 1.08)	0.359	0.74	(0.53, 1.03)	0.070	0.76	(0.54, 1.05)	0.098
Q3 (50.483–59.475)	0.79	(0.69, 0.91)	0.002	0.91	(0.78, 1.07)	0.248	0.58	(0.41, 0.80)	0.001	0.66	(0.47, 0.93)	0.019
Q4 (≥59.475)	0.64	(0.55, 0.74)	<0.001	0.78	(0.67, 0.91)	0.002	0.37	(0.26, 0.55)	<0.001	0.46	(0.30, 0.71)	<0.001
*p* for trend			<0.001			<0.001			<0.001			<0.001
**DII**												
Q1 (<−0.019)	**Reference**			**Reference**			**Reference**			**Reference**		
Q2 (−0.019–1.337)	1.24	(1.06, 1.44)	0.008	1.14	(0.97, 1.34)	0.111	1.45	(0.96, 2.20)	0.077	1.25	(0.83, 1.88)	0.293
Q3 (1.337–2.482)	1.45	(1.26, 1.66)	<0.001	1.26	(1.08, 1.46)	0.003	2.45	(1.69, 3.57)	<0.001	2.04	(1.38, 3.03)	<0.001
Q4 (≥2.482)	1.91	(1.66, 2.19)	<0.001	1.56	(1.34, 1.82)	<0.001	2.98	(2.07, 4.28)	<0.001	2.28	(1.56, 3.33)	<0.001
*p* for trend			<0.001			<0.001			<0.001			<0.001
**aMed**												
Q1 (<5)	**Reference**			**Reference**			**Reference**			**Reference**		
Q2 (5–5.5)	0.94	(0.82, 1.08)	0.385	1.01	(0.88, 1.17)	0.869	0.89	(0.61, 1.30)	0.547	0.92	(0.62, 1.38)	0.690
Q3 (5.5–6.5)	0.85	(0.74, 0.97)	0.018	0.94	(0.83, 1.07)	0.361	0.58	(0.43, 0.79)	<0.001	0.61	(0.44, 0.85)	0.004
Q4 (≥6.5)	0.64	(0.56, 0.74)	<0.001	0.81	(0.70, 0.93)	0.004	0.33	(0.22, 0.49)	<0.001	0.41	(0.26, 0.64)	<0.001
*p* for trend			<0.001			<0.001			<0.001			<0.001
**DASH**												
Q1 (<2.633)	**Reference**			**Reference**			**Reference**			**Reference**		
Q2 (2.633–3.405)	0.98	(0.83, 1.15)	0.784	1.03	(0.87, 1.22)	0.726	0.99	(0.72, 1.36)	0.938	1.01	(0.711.43)	0.964
Q3 (3.405–4.303)	0.89	(0.79, 1.00)	0.048	0.99	(0.87, 1.12)	0.840	0.88	(0.65, 1.21)	0.435	1.01	(0.71, 1.44)	0.940
Q4 (≥4.303)	0.82	(0.72, 0.94)	0.004	0.94	(0.81, 1.08)	0.366	0.63	(0.44, 0.88)	0.008	0.73	(0.50, 1.06)	0.096
*p* for trend			<0.001			0.163			0.012			0.170

Note: Model 1: Adjusted for age, gender, race/ethnicity; Model 2: Model 1 + body mass index + smoking status + poverty status + education levels + alcohol consumption + leisure time physical activity + history of diabetes. *p*-values less than 0.05 (*p*-value < 0.05) were considered significant. Abbreviations: CKD, chronic kidney disease; HEI, healthy eating index; DII, Dietary Inflammation Index; aMed, Alternate Mediterranean Diet Score; DASH, dietary approaches to stop hypertension index; OR, odd ratio; CI, confidence interval.

## Data Availability

NHANES: (https://www.cdc.gov/nchs/nhanes/index.htm); GWAS summary statistics: UK Biobank (https://www.nealelab.is/uk-biobank); FinnGen: (https://r9.finngen.fi).

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
