# Peer review of "The Combined Effects of the Most Important Dietary Patterns on the Incidence and Prevalence of Chronic Renal Failure: Results from the US National Health and Nutrition Examination Survey and Mendelian Analyses"

_nutrients, 2024, doi:10.3390/nu16142248_

Round 1

Reviewer 1 Report

Comments and Suggestions for Authors

I suggest reducing the manuscript's similarity percentage.

Was adjustment made for intrapersonal variance of nutrient and food values obtained from the 24-hour recall?

Review the section titled: 2.2.4. Diet (aMed) ScoreDietary Approaches to Stop Hypertension (DASH).

How were confounding factors chosen? Why was hypertension not included in the model?

Was any adjustment made for the multiple comparisons conducted?

Considering the number of results presented, the discussion is brief and limited.

Comments on the Quality of English Language

 Minor editing of English language required

Author Response

Comment 1:

I suggest reducing the manuscript's similarity percentage.

Response 1: Thank you for your suggestion and we have reduced the manuscript's similarity percentage in the manuscript.

Comment 2:

Was adjustment made for intrapersonal variance of nutrient and food values obtained from the 24-hour recall?

Response 2: Thank you for the comment. In our study, we adjusted for total energy intake in our model in order to balance individual dietary differences.

Review the section titled: 2.2.4. Diet (aMed) ScoreDietary Approaches to Stop Hypertension (DASH).

Comment 3:

How were confounding factors chosen? Why was hypertension not included in the model?

Response 3: Thank you for the comment. For the selection of confounding factors, we referred to two articles, Nutrients and BMC medicine, respectively, both of which were analysed using dietary data from the NHANES population. In this article “Dietary Quality and Relationships with Metabolic Dysfunction-Associated Fatty Liver Disease (MAFLD) among United States Adults, Results from NHANES 2017–2018”[1], researchers assessed the association between five dietary patterns, including DASH, DII, HEI, Med, and MAFLD adjusting for age, gender, race, education levels, PIR, physical activity levels, and smoking and drink conditions. In another article “Relationship between diet quality scores and the risk of frailty and mortality in adults across a wide age spectrum”[2], researchers assessed the association between five dietary patterns, including DASH, HEI, DII, Med, and the risk of mortality. The researchers adjusted for potential confounding factors (provided by NHANES) as underlying covariates, including age, sex, race, education level, marital status, employment status, smoking, body mass index, and study cohort. Synthesising the references, we included age, sex, race, education, PIR, smoking status, LTPA, BMI and history of diabetes as covariates.

We agree that hypertension is one of the risk factors for CKD. For your comment, we have provided three reasons. Firstly, referring to the article published in BMC medicine “The impact of non-alcoholic fatty liver disease and liver fibrosis on adverse clinical outcomes and mortality in patients with chronic kidney disease: a prospective cohort study using the UK Biobank”[3], we found that the researchers only adjusted for history of diabetes in patients with CKD and did not include blood pressure status in the model. Secondly, referring to the article published in Nutrients “Association of Dietary Intakes and Genetically Determined Serum Concentrations of Mono and Poly Unsaturated Fatty Acids on Chronic Kidney Disease: Insights from Dietary Analysis and Mendelian Randomization”[4], we found that essentially the same results were obtained regardless of whether hypertension was adjusted for in the model. Thirdly, systolic and diastolic blood pressures were measured in NHANES in only some of the participants. Therefore, if hypertension was put into the model, it would have reduced the number of participants entering the final analysis by a larger number.

Comment 4:

Was any adjustment made for the multiple comparisons conducted?

Response 4: Thank you for the comment. In the NHANES section, our study utilizes a public database for data mining analysis and is not designed as a confirmatory study. Therefore, it is not appropriate to apply multiple correction methods (such as the Bonferroni correction). In the MR section, to control the probability of Type I error, we considered the issue of multiplicity when selecting SNPs and chose a p-value threshold of <5*10^(-8).

Comment 5:

Considering the number of results presented, the discussion is brief and limited.

Response 5: Thank you for your suggestion and we discuss the results in more detail.

  • [1] Tian T, Zhang J, Xie W, Ni Y, Fang X, Liu M, Peng X, Wang J, Dai Y, Zhou Y. Dietary Quality and Relationships with Metabolic Dysfunction-Associated Fatty Liver Disease (MAFLD) among United States Adults, Results from NHANES 2017-2018. Nutrients. 2022 Oct 26;14(21):4505. doi: 10.3390/nu14214505.
  • [2] Jayanama K, Theou O, Godin J, Cahill L, Shivappa N, Hébert JR, Wirth MD, Park YM, Fung TT, Rockwood K. Relationship between diet quality scores and the risk of frailty and mortality in adults across a wide age spectrum. BMC Med. 2021 Mar 16;19(1):64. doi: 10.1186/s12916-021-01918-5.
  • [3] Hydes TJ, Kennedy OJ, Buchanan R, Cuthbertson DJ, Parkes J, Fraser SDS, Roderick P. The impact of non-alcoholic fatty liver disease and liver fibrosis on adverse clinical outcomes and mortality in patients with chronic kidney disease: a prospective cohort study using the UK Biobank. BMC Med. 2023 May 18;21(1):185. doi: 10.1186/s12916-023-02891-x.
  • [4] Mazidi M, Kengne AP, Siervo M, Kirwan R. Association of Dietary Intakes and Genetically Determined Serum Concentrations of Mono and Poly Unsaturated Fatty Acids on Chronic Kidney Disease: Insights from Dietary Analysis and Mendelian Randomization. Nutrients. 2022 Mar 15;14(6):1231. doi: 10.3390/nu14061231.

Reviewer 2 Report

Comments and Suggestions for Authors

The manuscript is very interesting. The authors present very important information. The authors use a sufficient methodology that allows evaluating the objective proposed in the study. However, I have the following comments.

Comments:

1. In the title I suggest indicating the country of origin of the study.

2. Improve the writing of the objective of the study.

3. Replace "gm" with "g", table 1 (example: Total sugars, gm for Total sugars, g

4. Table 1. I suggest separating the general information of the group studied from the nutritional information.

5. I suggest carrying out a study that allows identifying the foods that provide the nutrients studied. Especially those associated with protective effects. Such as fruits and vegetables, legumes, dairy products or eggs.

6. Ifura 4 shows relevant information. However, it is very difficult to read and understand. I suggest separating into two figures. In addition, including a letter for each variable. Example. Male (A). The letters are very small.

7. I suggest reviewing the writing of the entire manuscript.

8. The introduction is good and well referenced. However, the discussion is insufficient and very general. For example, the authors do not discuss the nutrients evaluated, and the discussion regarding diet is very limited. This is a major flaw in the manuscript.

Comments on the Quality of English Language

I suggest reviewing the writing of the entire manuscript.

Author Response

Comment 1:

  1. In the title I suggest indicating the country of origin of the study.

Response 1: Thank you for your suggestion and we have adapted it in the title.

Comment 2:

  1. Improve the writing of the objective of the study.

Response 2: Thank you for your suggestion and we have improved the objective of the study. This study aimed to investigate the role of four healthy dietary patterns in the incidence of chronic kidney disease (CKD) using data from the National Health and Nutrition Examination Survey (NHANES). Additionally, the study employed Mendelian randomization (MR) analysis to explore the potential causal relationship between these dietary patterns and CKD.

Comment 3:

  1. Replace "gm" with "g", table 1 (example: Total sugars, gm for Total sugars, g

Response 3: Thank you for your suggestion and we have adapted it in the table 1 and table S3.

Comment 4:

  1. Table 1. I suggest separating the general information of the group studied from the nutritional information.

Response 4: Thank you for your suggestion and We have split Table 1 into Table 1 and Table S3.

Comment 5:

  1. I suggest carrying out a study that allows identifying the foods that provide the nutrients studied. Especially those associated with protective effects. Such as fruits and vegetables, legumes, dairy products or eggs.

Response 5: Thank you for your suggestion. Our study focuses specifically on investigating the association between four dietary patterns and chronic kidney disease (CKD). However, we will consider exploring the role of specific foods, such as fruits, vegetables, legumes, dairy products, and eggs, in a subsequent phase of our research to further elucidate their potential protective effects.

Comment 6:

  1. Ifura 4 shows relevant information. However, it is very difficult to read and understand. I suggest separating into two figures. In addition, including a letter for each variable. Example. Male (A). The letters are very small.

Response 6: Thank you for your suggestion and we have split Figure 1 into Figure 1 and Figure S4.

Comment 7:

  1. I suggest reviewing the writing of the entire manuscript.

Response 7: Thanks to your suggestions, we have reviewed the entire manuscript for revisions.

Comment 8:

  1. The introduction is good and well referenced. However, the discussion is insufficient and very general. For example, the authors do not discuss the nutrients evaluated, and the discussion regarding diet is very limited. This is a major flaw in the manuscript.

Response 8: Thank you for your suggestion and we discuss the results in more detail.

Reviewer 3 Report

Comments and Suggestions for Authors

This is an interesting, well done and well written paper. I agree wholeheartedly with the proposed conclusions.

Nevertheless I have some minor concerns:

row 127: Section 2.2.4 "Diet (aMed) Score", is maybe "aMed" to be erased?

row 149: "of specific nutrients " is to be erased;

row 164: what about previously smokers?

row 172: R4.2.3, definition and source are lacking;

row 192: "SNPs" abbreviations should be defined the first time;

row 209: eemployed has a redundant "e";

row 218: "the" is repeated twice;

row 235: the source (city and state) of the employed software is not stated;

row 241: "but" and "was" are redundant;

Table 1: DASH showed a "significant" difference, while the difference is zero;

Table 1: Magnesium in the third column is 1.2 mg, something seems wrong;

Figure 1: is unreadable;

row 301 (and row 378): what does "fortified" wine mean?

Between rows 374 and 375: what about the odd association between VB12 and the risk of dialysis?

Comments on the Quality of English Language

English seems fine and only minor corrections are necessary.

Author Response

Comment 1:

row 127: Section 2.2.4 "Diet (aMed) Score", is maybe "aMed" to be erased?

Response 1: Thank you for your suggestion and we have adapted it in the manuscript.

Comment 2:

row 149: "of specific nutrients " is to be erased;

Response 2: Thank you for your suggestion and we have adapted it in the manuscript.

Comment 3:

row 164: what about previously smokers?

Response 3: Thank you for the comment. In our study, we categorised smoking status into smokers and never smokers. Ever smokers were included in the category of smokers. This is because we were unable to obtain the quitting time of former smokers, and the harms of smoking are cumulative.

Comment 4:

row 172: R4.2.3, definition and source are lacking;

Response 4: Thank you for your suggestion and we have adapted it in the manuscript.

Comment 5:

row 192: "SNPs" abbreviations should be defined the first time;

Response 5: Thank you for your suggestion and we have adapted it in the manuscript.

Comment 6:

row 209: eemployed has a redundant "e";

Response 6: Thank you for your suggestion and we have adapted it in the manuscript.

Comment 7:

row 218: "the" is repeated twice;

Response 7: Thank you for your suggestion and we have adapted it in the manuscript.

Comment 8:

row 235: the source (city and state) of the employed software is not stated;

Response 8: Thank you for your suggestion and we have adapted it in the manuscript.

Comment 9:

row 241: "but" and "was" are redundant;

Response 9: Thank you for your suggestion and we have adapted it in the manuscript.

Comment 10:

Table 1: DASH showed a "significant" difference, while the difference is zero;

Response 10: Thank you for the comment. In both the CKD and Non-CKD groups, the DASH scores were 3.46260 and 3.51238, respectively, but were shown to be the same as 3.5 because of the retention of one decimal place. T-tests showed that there was a significant difference in the DASH scores between these two groups.

Comment 11:

Table 1: Magnesium in the third column is 1.2 mg, something seems wrong;

Response 11: Thank you for your suggestion and we have adapted it in the manuscript. Due to an input error, the magnesium intake of 1.2 in the original Table 1 was corrected to 263.5 in the current Table S2.

Comment 12:

Figure 1: is unreadable;

Response 12: Thank you for your suggestion and we have split Figure 1 into Figure 1 and Figure S4.

Comment 13:

row 301 (and row 378): what does "fortified" wine mean?

Response 13: Thank you for your suggestion. Our data were driven from UK Biobank, the dietary questionnaire of which exemplified fortified wine with Sherry, Port, Vermouth, Muscat, Madeira, Malaga, Tokay, Frontignan, Frontignac.

Comment 14:

Between rows 374 and 375: what about the odd association between VB12 and the risk of dialysis?

Response 14: Thank you for your suggestion and we have corrected the results in the manuscript.

Round 2

Reviewer 2 Report

Comments and Suggestions for Authors

Authors answered all my comments. Therefore, the manuscript can be accepted. 

Author Response

Comment 1:

Authors answered all my comments. Therefore, the manuscript can be accepted. 

Response 1:Thanks to your approval, we've made all the changes in the manuscript.